# A Mixture of Chemicals Found in Human Amniotic Fluid Disrupts Brain Gene Expression and Behavior in *Xenopus laevis*

**DOI:** 10.3390/ijms24032588

**Published:** 2023-01-30

**Authors:** Michelle Leemans, Petra Spirhanzlova, Stephan Couderq, Sébastien Le Mével, Alexis Grimaldi, Evelyne Duvernois-Berthet, Barbara Demeneix, Jean-Baptiste Fini

**Affiliations:** Département Adaptations du Vivant (AVIV), Physiologie Moléculaire et Adaptation (PhyMA UMR 7221 CNRS), Muséum National d’Histoire Naturelle, CNRS, CP 32, 7 rue Cuvier, 75005 Paris, France

**Keywords:** thyroid hormones, endocrine disruption, neurodevelopment, *Xenopus laevis*

## Abstract

Thyroid hormones (TH) are essential for normal brain development, influencing neural cell differentiation, migration, and synaptogenesis. Multiple endocrine-disrupting chemicals (EDCs) are found in the environment, raising concern for their potential effects on TH signaling and the consequences on neurodevelopment and behavior. While most research on EDCs investigates the effects of individual chemicals, human health may be adversely affected by a mixture of chemicals. The potential consequences of EDC exposure on human health are far-reaching and include problems with immune function, reproductive health, and neurological development. We hypothesized that embryonic exposure to a mixture of chemicals (containing phenols, phthalates, pesticides, heavy metals, and perfluorinated, polychlorinated, and polybrominated compounds) identified as commonly found in the human amniotic fluid could lead to altered brain development. We assessed its effect on TH signaling and neurodevelopment in an amphibian model (*Xenopus laevis*) highly sensitive to thyroid disruption. Fertilized eggs were exposed for eight days to either TH (thyroxine, T_4_ 10 nM) or the amniotic mixture (at the actual concentration) until reaching stage NF47, where we analyzed gene expression in the brains of exposed tadpoles using both RT-qPCR and RNA sequencing. The results indicate that whilst some overlap on TH-dependent genes exists, T_4_ and the mixture have different gene signatures. Immunohistochemistry showed increased proliferation in the brains of T_4_-treated animals, whereas no difference was observed for the amniotic mixture. Further, we demonstrated diminished tadpoles’ motility in response to T_4_ and mixture exposure. As the individual chemicals composing the mixture are considered safe, these results highlight the importance of examining the effects of mixtures to improve risk assessment.

## 1. Introduction

In 1894, two clinical studies demonstrated that some features of mental retardation characteristic of cretinism could be improved by treating young patients with thyroid extract [1,2]. Since then, the role of TH in brain maturation has been documented in detail [3]. Most research focuses on the rapid period of brain growth, i.e., the perinatal period in mammals [3,4] Recent epidemiological and clinical studies have suggested that the early stages of neurogenesis are also TH-dependent [5,6,7]. The main idea arising from these studies is that before the development of the fetal thyroid (which is not functional before the 16th week of gestation in humans [8]), maternal TH levels are critical in determining postnatal neuro-motor development of the child. Various studies have shown that a child’s IQ is correlated with their mother’s thyroid status during early pregnancy [6,9].

Experimental studies on the role of functional thyroid signaling during early embryogenesis are more easily addressed in free-living embryos that do not require dissection from the mother. In the *Xenopus laevis* embryo, the thyroid gland is observed at NF (Nieuwkoop and Faber) 43 [10], which is at about one week of age in standard rearing conditions. Analyzing morphological and biochemical data as well as thyroid receptor alpha (*thra*) mRNA profiles led to the proposition that *Xenopus laevis* embryos start to show competence to respond to T_3_ treatment between stages NF40-44 [4,11,12,13]. Our research team previously demonstrated that exposing tadpoles to TH at stage NF 37 for 24 h induced TH-regulated gene response in brain tissue at stage NF 41 [14].

Multiple studies have documented significant contamination of human populations and wildlife by multiple anthropogenic chemicals [15,16]. About 30 anthropogenic chemicals are present in all American women, with 15 being ubiquitous, including in pregnant women [15]. Most of these chemicals are demonstrated or suspected TH disruptors [17,18], raising the question of whether current exposure to ubiquitous chemicals affects thyroid signaling and thereby early brain development. Even though certain xenobiotics have been investigated for their actions on specific endocrine axes, few studies have addressed their combined or ‘cocktail’ effects. We previously examined the consequences of a 3-day exposure to a mixture made of 15 ubiquitous molecules [19]. The results suggested a T_3_-like effect on most of the endpoints measured. However, some gene expression patterns in brain tissue revealed the opposite effects when comparing T_3_ or mixture-induced effects.

A growing body of evidence demonstrates that the entire duration of pregnancy is a sensitive window for toxicant exposure. Therefore, we investigated a more extended exposure period (8 days) to the mixture (see Table 1), as reported by Fini et al. [19], at environmentally relevant concentrations during embryogenesis, further referred to as the 1x concentrated mixture. After exposure, gene expression analysis was conducted employing RT-qPCR and RNA sequencing on the brains of *Xenopus laevis* tadpoles exposed during the embryonic period. Additionally, immunohistochemistry (IHC) was performed on brain samples to unravel possible aberrant molecular mechanisms of neurodevelopment after mixture exposure. Furthermore, the behavior of tadpoles was analyzed, as altered behavior could be a possible marker of abnormal neurodevelopment. A positive control, TH (T4, 10 nM), was included in all the experiments. No aberrant survival rates, growth effects or potential malformations were observed for the animals exposed to the 1x mixture concentration. TH-exposed animals had a slightly lower survival rate and a higher incidence of malformation compared to the control group after the exposure.

## 2. Results

Exposure to the amniotic mixture or THs alters the expression of genes essential in the TH signaling pathway and affects neuronal developmental genes.

A schematic representation of the conducted exposure study is depicted in Figure 1. Embryonic mixture and TH exposure (T_3_ 5 nM and T_4_ 10 nM) were tested to evaluate their effects on gene expression in brain tissue. First, a selection of genes was elected, to be measured with RT-qPCR, based on their involvement in the TH signaling pathway and their essential role in normal brain development.

The expressions of TH-dependent transcription factors, Kruppel-like factor (*klf9*) (Figure 2E), and TH receptor beta (*thrb*) (Figure 2J) were induced after the T_3_, T_4_, and amnio mixture exposure. The TH receptor alpha (*thra*) (Figure 2I) was only up-regulated under mixture exposure. Further, the expression of membrane transporters allowing specific TH transporters *mct8* (Figure 2G) and *oatp1c1* (Figure 2F) was investigated. Interestingly, while both *mct8* and *oatp1c1* were up-regulated in mixture-exposed brains, the expression of *mct8* was down-regulated by T_3_ and T_4_, showing an opposite expression pattern. Under both T_3_ and T_4_ exposure, the expression of deiodinase 1 (*dio1*) was down-regulated and the expression of deiodinase 3 (*dio3*) was up-regulated to maintain TH homeostasis. Under mixture exposure, deiodinase 2 (*dio2*) was induced, similar to the T_4_ exposure. Furthermore, the effects on genes that are dysregulated in human autism spectrum disorder patients were investigated (e.g., *sin3a*, *bdnf*, *mbp*, and *mecp2*). Strikingly, all neurodevelopmental genes were found to be induced after mixture exposure and found to be reduced after TH exposure (Figure 2D,H,K,L).

Taken together, a remarkable impact of the amniotic mixture on brain gene expression was observed by RT-qPCR, and promoted us to investigate possible effects on a much broader range of genes. Given the multiple interesting opposite TH effects compared to the mixture effect, we performed RNA sequencing on samples of both TH (T_4_ 10 nM) and amniotic-treated animals.

A principal component analysis (PCA) was conducted for all the sequenced brain samples using their respective gene expression profiles for their representation on a two-dimensional graphic (Figure 3A). Each dot in the PCA graphic represents the gene expression profile of a brain sample (with each sample containing a pool of two brains), and the distance between two dots is proportional to the extent of similarity between the gene expression profiles. The PCA plot containing a projection on the first two principal components, which together explain 60.24% (49.70% + 10.54%) of the total variance, illustrated that the exposed samples clustered apart from the control group. The RNA sequencing analysis revealed that 587 transcripts were significantly altered in the group exposed to the amniotic 1x mixture compared to the controls, whereas 6552 were significantly altered after exposure to T_4_. A total of 210 genes were commonly found to be affected by both TH or mixture exposure (Figure 3B). Of the differentially expressed genes (DEGs) from the mixture exposure, more than 70% appear to be down-regulated (Figure 3C). In the case of TH exposure, many genes were significantly down- or up-regulated (Figure 3C, D).

KEGG pathway analysis indicated that the predicted targets of mixture exposure are involved in biological processes such as the MAPK signaling pathway (Table 2). This pathway is connected with the cell cycle and particularly the G1/S switch [20]. The second pathway we found to be enriched by the amniotic mixture in the embryonic brain is lysosomal activation. This pathway is considered to be a milestone in several cell death pathways, including apoptosis [21]. Another target of TH exposure was the interaction between various neuroactive ligands and receptors (Table 2).

Given the 35% overlap of DEGs between mixture-treated and TH-treated genes, we further investigated the commonality between these two exposure conditions by crossing them with a dataset containing TH-responsive genes obtained from Chatonnet et al. (2015) [22]. Twelve genes are at the intersection of these three lists (DEGs amnios, DEGs T_4_, and TH-responsive genes) (Figure 4A), nine of which are regulated in the same direction. Interestingly, *tshb*, a gene encoding for thyrotropin, a thyroid-stimulating hormone that induces the thyroid gland to produce T_4_, is up-regulated in the amniotic mixture.

The amniotic exposure affected the MAPK signaling pathway, and given the importance of this signaling pathway in learning and memory and its implication in neurodevelopmental disorders [23,24,25,26] we crossed the DEGs of amniotic exposure with the SFARI dataset, an online database of autism genes. Eighteen genes are at the intersection with the SFARI database: half of them are also shared with the T_4_ DEGs (Figure 4B). Altered neurodevelopmental important RNAseq transcripts were confirmed by employing RT-qPCR (Figure 5).

Given the crucial role of TH in both proliferation and apoptosis and the predicted effects of the amniotic mixture on both the MAPK signaling pathway and lysosomal activation, we investigated the proliferation/cell death ratios. Following an 8-day exposure, we fixed the tadpoles and performed immunohistochemistry on cryosections on heads of exposed tadpoles, using anti-phosphorylated histone H3 (P-H3), a mitotic marker, and anti-caspase 3 as an apoptotic marker. We measured an increase in proliferating cells (PH3+ cells) after T_4_ exposure but not with the amniotic mixture. No significant difference was detected for the apoptotic marker (Figure 6). Results obtained from animals treated with T_4_ confirm the previously demonstrated increase in proliferation after TH treatment and served as a positive control [27].

### Exposure to an Amniotic Mixture Alters the Tadpole Behavior

Next, we addressed the phenotypic consequences of early exposure, since motor behavior changes can imply an alteration in the neural circuitry controlling movement [28]. For this, we used a video tracking system and recorded the total distance traveled by individual tadpoles with 30 s, alternating light and dark cycles for a total of 10 min.

The distance traveled decreased with the mixture and TH exposure in the light periods by 32% on average for the 1x-mixture-treated animals and by 38% for the TH-treated animals (Figure 7A,B).

## 3. Discussion

The dependence of amphibian metamorphosis on T_3_ [29] has led to the wide use of *X. laevis* for analyzing TH action and interference with TH signaling. Given the central role of TH in orchestrating metamorphosis, initial work on TH levels and amphibian development focused on measuring components of the TH signaling pathways from NF 54 (prometamorphosis) onwards [30]. However, more recent work has shown that eggs and embryos of fish, reptiles, birds [31,32,33], and amphibians [15,34] contain both tetraiodothyronine, T_4_, and the more biologically active form of the hormone, triiodothyronine, T_3_. We questioned the impact of external exposure to TH, from fertilized eggs up until the developing larvae (NF1-NF47). In the series of experiments that we did not present in this manuscript, we discovered that survival was affected by all TH antagonists tested, i.e., NH3 (10^−7^ M), iopanoic acid (IOP 10^−6^ M), and methimazole (MMI, 10^−4^ M), either during the 8-day treatment or the 4 days after it. All animals treated with NH3 died, and only 5% of the animals exposed to MMI could reach metamorphosis. Interestingly, the T_3_ 5 nM treatment also induced mortality just a few days after the end of the treatment.

We questioned whether “less-toxic” treatments such as T_4_ 10 nM (a concentration equivalent to those found during metamorphosis) or an amniotic mixture replicated from a mixture of 15 ubiquitous compounds at concentrations measured in human amniotic fluid [16], would modify brain gene expression in tadpoles exposed to fertilization through embryogenesis. Through examining effects on well-characterized direct T_3_ target genes, notably *klf9* and *TRβ*, we show that T_4_ and the amniotic mix exhibited similar effects, suggesting a pro-thyroid effect. In our previous study using the same mixture [1], during a shorter exposure time (3 days) at 1-week post fertilization (NF 45–NF 47), we observed increased GFP signaling using the Xenopus Eleutheroembryonic Thyroid Assay (XETA). Results included a loss of mobility and increased proliferation in neurogenic zones, also suggesting a T3-like effect, at least in some cells. However, in this manuscript, we used a non-biased approach for candidate gene research using RNA sequencing after embryonic exposure (NF 1–NF 47).

The KEGG pathway analysis identified up-regulated or down-regulated DEGs in numerous pathways following T_4_ treatment, including a gene set of neuroactive ligand–receptor interactions (Table 2). This list contains a series of altered genes affecting a myriad of downstream target genes and physiological functions, through direct pathways or via cross-talk between these genes coding for various neuropeptides, hormones, and neurotransmitters [35]. While predicted KEGG pathways affected by the amniotic mixture suggested possible effects on apoptosis and proliferation, immunohistochemistry only revealed an increase in proliferative cells in the region analyzed (the midbrain) in the group exposed to T_4_. However, this does not exclude the possibility of effects on proliferation and/or apoptosis in more specific regions of the brain.

Interestingly we found that the *thsb* gene, encoding the pituitary hormone thyrotropin, was significantly deregulated, in opposite ways by treatment of T_4_ or the amniotic mixture. Circulating TSH activates the synthesis of THs in the thyroid and represses TRH (thyrotropin-releasing hormone) at the hypothalamus level (CRH (corticotropin-releasing hormone) in amphibians, [36]). This circulating hormone activates the synthesis of TH in the thyroid gland and is measured at birth in humans for blood spot tests to analyze both TSH levels and different aspects of TH regulation. Higher TSH levels are indicative of problems in the thyroid axis, including the fact that a normal postnatal peak of TH has not occurred [37]. Here, we observe a very strong reduction in the expression of *tshb* after treatment of T_4_ at 10 nM, suggesting a centrally compensated hyperthyroid state. In zebrafish, Tonyushkina et al. [38] observed that *tshb* transcription was effective 96 h post fertilization. They also showed that *tshb* and *dio2* were co-expressed in thyrotrope cells and that their numbers were reduced after T_4_ treatment. Here, we observe that T_4_ treatment reduced expression of *tshb*, increased *dio3* expression, and decreased *mct8* expression, suggesting a feedback mechanism in response to an excess of TH in the brain. These findings demonstrate that even when the thyroid gland is not yet synthetizing TH, brain cells are able to cope with an excess of T_4_.

Strikingly, the treatment with the amniotic mixture leads to an opposite response, with overexpression of the *tshb* gene, increased expressed of TH transporters *oatp1c1* and *mct8*, and increased *dio2* expression, suggesting a hypothyroid state being compensated for centrally. The increase in *dio2* expression observed after T_4_ treatment in RNA sequencing and not in qPCR is not well understood. *Dio2* converts T_4_ into T_3_ and therefore contributes to the TH excess, and it cannot be ruled out that *dio2* overexpression could also convert T_3_ into T_2_, documented as less active than T_3_ [39].

Two things stand out from this result: firstly, the mixture has an anti-thyroid effect as well as some T_3_-like effects, even though some parameters point in the same direction (notably behavior), and secondly, tadpoles can be affected by a suggested hypothyroidism at the stage when the thyroid gland is formed but only just beginning to synthesize TH. These results highlight that peripheral mechanisms of deiodination are crucial during early development. Activating and inactivating deiodinases are intimately involved in determining T_3_ availability in specific tissues and cell types, and inter-ring deiodination (IRD) and outer-ring deiodination (ORD) are also present at these early embryonic stages mostly in the head region [40,41].

Our findings illustrate the concept that too little or too much TH during critical phases of development can cause adverse effects. Notably, results from human epidemiology show that low or high maternal TH, or low iodine during early pregnancy, is associated with a loss of IQ and impacts cortical thickness [9]. It is even more striking that Korevaars’ results showed that the variations in maternal TH were within normal levels.

To conclude, we have shown that early exposure to a mixture of EDCs commonly found in the human amniotic fluid may induce adverse effects as a T_4_ treatment. This latter condition is equivalent to maternal hyperthyroidism in humans. It is remarkable that both treatments disrupt *tshb*, brain gene expression, and result in altered behavior. Specifically, even though exposure to T_4_ or the amniotic mixture may result in opposite thyroid status, effects appear identical. From a molecular point of view, the majority of the common DEGs are regulated in the same direction, and from a macroscopic point of view, a loss of mobility is measured in both cases. These findings highlight the need for specific endpoint markers of TH disruption. The obtained results call urgent attention to the necessity of enhanced protection of humans and the environment from EDCs, particularly those that affect the thyroid axis.

## 4. Material and Methods

### 4.1. Chemical Exposure

The 10,000× exposure solution was prepared according to Fini et al. [19]. The 1x exposure solution was prepared by adding 1 µL of 10,000× concentrated mixture to 10 mL of Evian water. In all experiments, stage NF1 *Xenopus laevis* eggs (dejellied by cysteine) were placed into 6-well plates (15 tadpoles per well). An amount of 8 mL of previously prepared 1× exposure solution was added into the corresponding well after the removal of any excess liquid. The final concentration of DMSO was 0.01% in all groups, including the control. Multi-well plates were kept in the dark incubator at 23 °C for eight days to prevent chemical degradation related with light exposure. The renewal was daily, exposure solutions being prepared extemporaneously for eight days. For each endpoint, different amounts of technical replicates were conducted, which are noted under the corresponding paragraph.

### 4.2. RNA Extraction

At the end of the chemical exposure described above, tadpoles were anesthetized in 0.01% MS-222, and their brains were dissected under sterile conditions. Two brains were placed in 1.5 mL tubes containing 100 µL of lysis solution from an RNAqueous –micro kit (Thermofisher, Waltham, MA, USA) and flash-frozen in liquid nitrogen followed by storage at −80 °C. In total, five tubes containing two brains each per exposure group and per replicate were collected. RNA extraction was performed using the RNAqueous–micro kit following the manufacturer’s instructions. RNA concentrations were measured by a NanoDrop spectrophotometer (ThermoScientific, Rockford, IL, USA) and RNA quality was verified by BioAnalyzer (Agilent Technologies, Santa Clara, CA, USA). Only samples with an RIN > 7 were selected for further study. cDNA was synthesized using Reverse Transcription Master Mix (Fluidigm Corporation, San Francisco, CA, USA).

### 4.3. RT-qPCR

cDNA was diluted 1/20 in nuclease-free water. A quantitative PCR reaction was performed in 384 well-plates, with a standard reaction containing 1 μL of cDNA and 5 μL of the mix (3 µL of Power SYBR master mix, 1.7 µL of nuclease-free water, 0.15 µL of reverse primer (10 pM) and 0.15 µL of forward primer (10 pM) per well). The measurement was carried out by the QuantStudio 6 Flex (Life technologies, Carlsbad, CA, USA) device. The 2^−ΔΔCt^ method was used to calculate the relative concentrations of cDNA for the analysis of relative changes in gene expression (see [42] for a detailed description). The geometric mean of endogenous controls *ralb* and *ube2m* was used for normalization.

Data are represented as fold change (2^−ΔΔCt^) using a log (base2) scale plotted as a traditional box, and whisker plot by Tukey where the bottom and top of the box represent the 25th lower and 75th percentile, and the median is the horizontal bar in the box. Statistical analyses were performed on delta Cts using non-parametric Mann–Whitney in case of non-normal distributed data. In the case of a normal distribution, a Student t-test was conducted. Significance was determined at *p* < 0.05 (*), *p* < 0.01 (**), and *p* < 0.001 (***). Results are a pool of 6 independent replicates with 5 samples by group by replicate. Each sample consists of a pool of brains originating from two distinct tadpoles.

### 4.4. RNA-Sequencing

Two sets of experiments were realized at two different time points. The transcriptome libraries were prepared from total RNA using Illumina TruSeq Stranded mRNA Sample Preparation kits (Illumina Inc., San Diego, CA, USA). The libraries were sequenced on the Illumina NextSeq device using the 75bp single-end sequencing strategy with the TruSeq kit (Illumina Inc., San Diego, CA, USA).

Raw reads were first cleaned by removing PCR bias conserving one copy per cluster of duplicated reads (python script). Then, reads were trimmed for the first 11 and last 1 nucleotides (fastx-trimmer from fastx-toolkit version 0.0.13.2) to remove adapter remnants. Reads 64bp long were specifically selected (cutadapt version 1.15). The global quality of the reads was checked using the FastQC (version 0.11.2). Bowtie2 (v2.2.4) was used to map the clean reads against the *Xenopus laevis* genome (release 9.2) downloaded from XenBase (https://ftp.xenbase.org/pub/Genomics/JGI/Xenla9.2/, accessed on 18 April 2019) only conserving uniquely mapped reads. Reads counted on gene annotations was accomplished by HTSeq-count (v0.9.1) in union mode against the annotation of *Xenopus laevis* genome downloaded from XenBase (https://ftp.xenbase.org/pub/Genomics/JGI/Xenla9.2/XENLA_9.2_Xenbase.gff3, accessed on 18 April 2019).

Raw read counts were normalized by the variance stabilization transformation of DESeq (R package—version 1.16.1) and used to check the global behavior of the libraries by a principal components analysis (R package FactoMineR—version 1.39). This revealed a batch effect between samples on component 2 of the PCA. We chose to remove the complete impact of component 2 by recalculating a denoised table of raw read counts (R scripts). This consists in calculating the variance stabilizing matrix from the raw read counts table, performing the principal components analysis with the package prcomp, initializing the variable loading of interest (component 2) at zero, and reconstructing the denoised raw read counts matrix. Then, this new denoised table was used by DESeq for the differential expression analysis. Four different samples were employed coming from two technical replicates (two different breedings). Each sample contains a pool of brains originating from two distinct tadpoles.

### 4.5. DAVID

Differentially expressed genes were submitted to DAVID database (https://david.ncifcrf.gov/, accessed on 19 January 2023) for systematically extracting biological meaning for them by retrieving pathway maps from the Kyoto Encyclopedia of Genes and Genomes (KEGG).

### 4.6. Mobility

After the eight days of chemical exposure from NF 1 to NF 47, tadpoles were rinsed, placed separately into a 12-well plate containing 4 mL of Evian water in each well, and left to accommodate for 15 min. The behavior of the tadpoles was recorded by DanioVision (11.5, Noldus, Wageningen, The Netherlands)) behavior analysis system during a 10 min trial composed of ten 30 s light-on/30 s light-off intervals. During the light-on phase, maximal light stimulus (5 K Lux) was set. The total distance traveled by each tadpole was calculated by EthoVision software XT (11.5). The mean distance traveled by the control group in the 0–10 s period of the first light-on interval was used to normalize all data. Statistics were conducted using multiple t-tests with FDR approach: FDR (Q) = 5%, * *p* < 0.05, ** *p* < 0.01, *** *p* < 0.001, **** *p* < 0.0001. For each exposure, 3 technical replicates were conducted. These replicates were pooled and each replicate contained between 37 and 48 tadpoles per condition. The total amount of tadpoles used can be found within the figure legend.

### 4.7. Immunohistochemistry

Tadpoles were euthanized in MS-222 1 g/L for 15 min. Whole tadpoles were fixed in 4% paraformaldehyde overnight at 4 °C under agitation and transferred for storage in PFA 0.4%. The night before embedding in an optimal cutting temperature compound, samples were cryoprotected in sucrose 15% (in PBS) at 4 °C. Coronal cryosections (12 μm thickness) of whole tadpoles spanned the region containing lateral ventricles to the mid-hindbrain region. For immunohistochemical investigation of sections on slides, the following antibodies were used: (1) primary antibodies anti-Ser10 phosphorylated on Histone H3 mouse (05–806 Millipore) at 1/300 dilution and anti-caspase3 rabbit (Ab 3022–Abcam) at 1/200 dilution; (2) secondary antibodies Alexa Fluor 488 anti-mouse (A11029 Invitrogen) and Alexa Fluor 594 (A11012 Invitrogen) at 1/500 dilution. Image analysis was conducted in the midbrain (delimited by the end of lateral ventricles and the end of the optic tectum). All positive nuclei were manually counted from 2 independent experiments with n = 2 tadpoles per experiment, and statistical analyses were performed using the non-parametric Mann–Whitney test compared to the control group. Three technical replicates (3 individual breedings) were used wherefrom two tadpoles per breeding were analyzed.

## Figures and Tables

**Figure 1 ijms-24-02588-f001:**
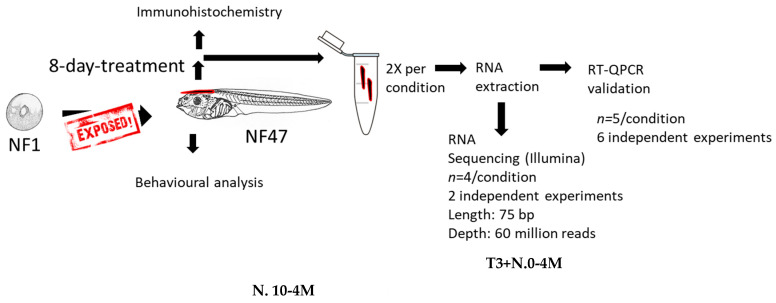
Schematic representation of the conducted exposure study. Fifteen *X. laevis* tadpoles per group were exposed for 8 days starting just after fertilization (NF 1) to stage NF 47 before gene expression analysis, mobility assays, and brain immunohistochemistry.

**Figure 2 ijms-24-02588-f002:**
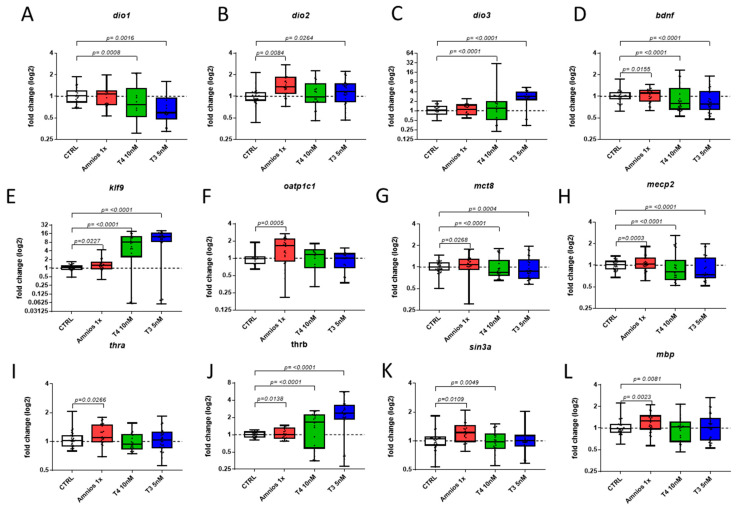
Gene expression after embryonic exposure to the amniotic mixture and THs. Embryos (NF 1–NF 47) were exposed to DMSO (CTRL), amniotic mixture at 1x concentration (Amnios 1X), T_4_ (10 nanomolar), and T_3_ (5 nanomolar). After brain dissection at stage NF 47, RNA extraction and RT-qPCR were conducted on genes involved in the TH signaling pathway, (**A**) *dio1*, (**B**) *dio2*, (**C**) *dio3*, (**E**) *klf9*, (**F**) *oatp1c1*, (**G**) *mct8*, (**I**) *thra*, (**J**) *thrb*, and brain development, (**D**) *bdnf*, (**H**) *mecp2*, (**K**) *sin3a*, and (**L**) *mbp*. Results are normalized to the geometric mean of the expression levels of the genes *ube2m* and *ralb*. Results are a pool of 6 independent replicates with *n* = 5 by group by replicate. The line in each box represents the median.

**Figure 3 ijms-24-02588-f003:**
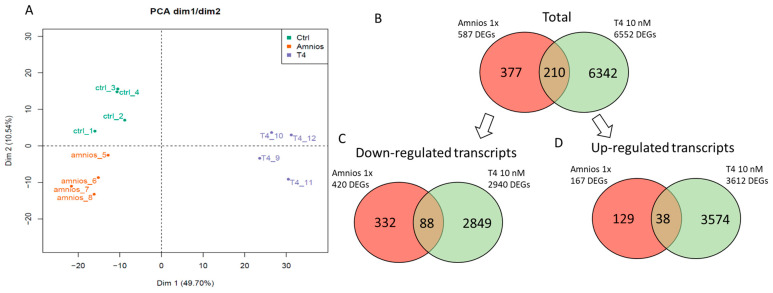
Principal component analysis plot of gene expression profiles from brain samples of exposed Xenopus tadpoles and Venn diagram from differentially expressed transcripts. (**A**) The profiles from the amniotic mixture (amnios (*n* = 4)) cluster separately to clusters representative of T4 (T4 10 nM (*n* = 4)) or control exposures (control (*n* = 4)). (**B**) Venn diagram showing the differentially expressed genes in each group compared to the control. (**C**) Venn diagrams showing either down- or (**D**) up-regulated genes compared to the controls.

**Figure 4 ijms-24-02588-f004:**
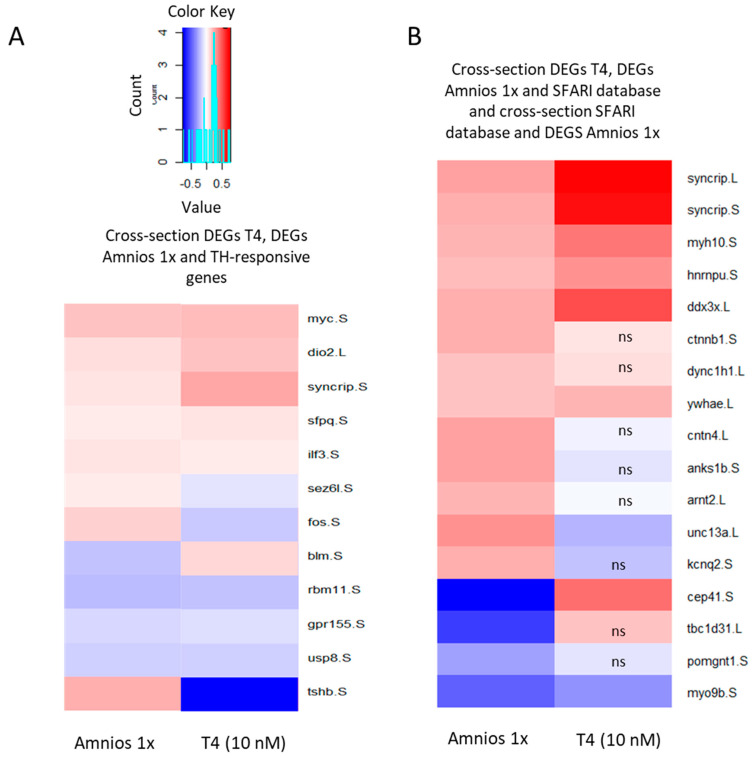
Changes in expression levels of genes specifically affected by amniotic mixture or T_4_ treatment. Heatmap depicts the average change in the expression level of genes affected by either amniotic mixture (Amnios 1X) or T_4_ (10 nM). Gene names are shown at the right of the heatmap. The color bar represents log2 differences from the control for each treatment. (**A**) DEGs that are in common between T_4_ and amniotic treatment as well as TH-responsive genes [22]. (**B**) DEGs that are common between either amniotic mixture and SFARI gene list and DEGs in common between Amnios 1X, T_4_ (10 nM), and SFARI gene list. The letters ns stand for non-significant; these genes do not belong to the DEG list.

**Figure 5 ijms-24-02588-f005:**
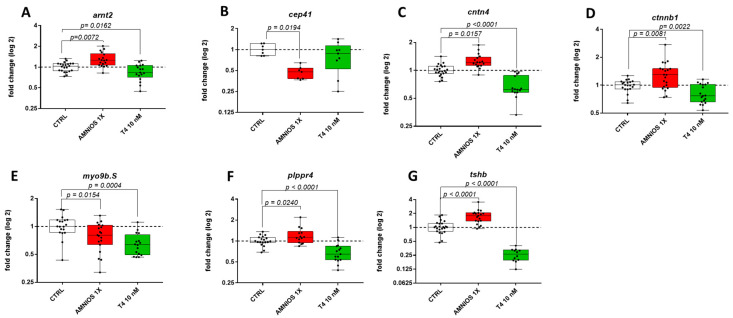
Gene expression after embryonic exposure to the amniotic mixture and THs. Embryos (NF 1–NF 47) were exposed to DMSO (CTRL), amniotic mixture at 1x concentration (Amnios 1X), and T_4_ (10 nanomolar). After brain dissection at stage NF 47, RNA extraction and RT-qPCR were conducted on genes involved in TH signaling or autism-related genes, (**A**) arnt2, (**B**) cep41, (**C**) cntn4, (**D**) ctnnb1, (**E**) myo9b, (**F**) plppr4, (**G**) tshb. 2.1. Exposure to TH but Not Amniotic Mixture Induces Both Apoptosis and Proliferation.

**Figure 6 ijms-24-02588-f006:**
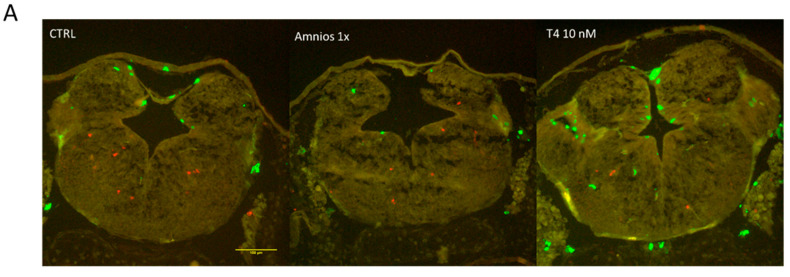
Effect of Amnios 1X and T_4_ on proliferation and apoptosis in tadpole brains. For each treatment group, 2 tadpoles from 3 different females were analyzed. (**A**) Example cross-sectional images of brains following immunochemistry: apoptosis marker in red (caspase3), proliferation marker in green (PH3). Scale bar, 100 µm. (**B**) Average number of positive cell nuclei per section for caspase3 and PH3, and the ratio of proliferative/apoptotic cells.

**Figure 7 ijms-24-02588-f007:**
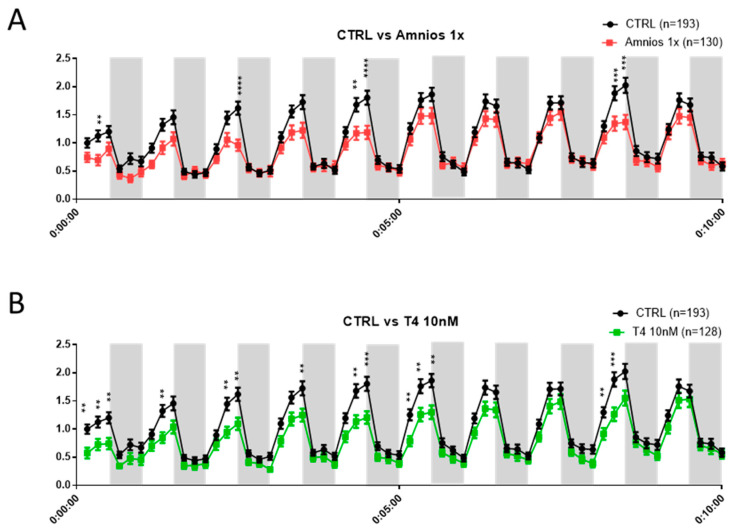
Behavioral study of tadpoles exposed to amniotic mixture and T_4_ (10 nM). The normalized distance was measured during 10 min trials with 30 s light/30 s dark alternation using Video tracking Noldus Ethovision system. NF 47 tadpoles, directly after 8-day exposure to either (**A**) the amniotic mixture or (**B**) T_4_ (10 nM), were used to investigate traveled distance. Graph: mean +/− SEM, multiple t-tests with FDR approach, FDR (Q) = 5%, ** *p* < 0.01, *** *p* < 0.001, and **** *p* < 0.0001. For each exposure, 3 technical replicates were conducted. These replicates were pooled, and each replicate contains between 37 and 48 tadpoles per condition. The total amount of tadpoles used can be found within the figure legend.

**Table 1 ijms-24-02588-t001:** Composition of chemical mixture.

n°	Family	Molecule	Concentration 1X (Actual Concentration Found in Amniotic Fluid)
1	Phenol	Bisphenol A	0.2 × 10^−8^ M
2	Phenol	Triclosan	0.7 × 10^−7^ M
3	Phenol	Benzophenone-3	0.86 × 10^−7^ M
4	Phthalate	Dibutyl phthalate	0.24 × 10^−6^ M
5	Phthalate	Di-2-ethylhexyl phthalate	0.1 × 10^−6^ M
6	Organochlorine pesticide	Hexachlorobenzene	0.8 × 10^−11^ M
7	Organochlorine pesticide	Dichlorodiphenyldichloroethylene	0.66 × 10^−9^ M
8	Perfluorinated compound	Perfluorooctanoic acid	0.43 × 10^−8^ M
9	Perfluorinated compound	Perfluorooctanesulfonic acid	0.8 × 10^−8^ M
10	Poly aromatic hydroxylated compound	2-napthol	0.5 × 10^−8^ M
11	Polychlorinated compound	Sodium perchlorate monohydrate	0.3 × 10^−8^ M
12	Polybrominated compound	Decabromodiphenyl ether	0.63 × 10^−9^ M
13	Polychlorinated compound	PCB-153	0.2 × 10^−8^ M
14	Heavy metal	Methyl mercury(III) chloride	0.5 × 10^−7^ M
15	Heavy metal	Lead (II) chloride	0.21 × 10^−8^ M

**Table 2 ijms-24-02588-t002:** KEGG pathway analysis of target genes after mixture and TH exposure. Target genes up-regulated or down-regulated that were annotated for neuroactive ligand–receptor interaction under T_4_ (10 nM) exposure, and the MAPK signaling pathway and lysosome activity under mixture exposure (Amnios 1X).

Exposure	Gene		Pathway
**Mixture 1x**	cacnb1.S (calcium channel, voltage-dependent, beta 1 subunit S) nfkb2.S (nuclear factor of kappa light polypeptide gene enhancer in B-cells 2 (p49/p100) S) myd88.S (myeloid differentiation primary response 88 S) irak4.L (interleukin 1 receptor associated kinase 4 L) fos.S (FBJ murine osteosarcoma viral oncogene S) myc.S (v-myc avian myelocytomatosis viral oncogene S) hspa8.L (heat shock protein family A (Hsp70) member 8)		MAPK signaling pathway
**Mixture 1x**	ctns.L (cystinosin, lysosomal cystine transporter glb1l.L (galactosidase beta 1 like L) naga.L (N-acetylgalactosaminidase, alpha- L) mfsd8.L (major facilitator superfamily domain containing 8 L) galc.L (galactosylceramidase L) arsa.1.S (arylsulfatase A, gene 1 S) ap3s2.S (adaptor related protein complex 3 sigma 2 subunit S) gm2a.L (GM2 ganglioside activator L)		Lysosome
**T4 (10 nM)**	gria1.L/S (glutamate receptor, ionotropic, AMPA 1 L/S) kiss1.S (kisspeptin S) tshb.L.S (thyroid stimulating hormone, beta L/S) gabrd.L (gamma-aminobutyric acid (GABA) A receptor, delta L) gnrh2.L (gonadotropin releasing hormone 2 L) htr2C.L (5-hydroxytryptamine (serotonin) receptor 2C, G protein-coupled L) glrb.L/S (glycine receptor beta L/S) gabbr1.S (gamma-aminobutyric acid (GABA) B receptor, 1 S) pth2R.L (parathyroid hormone 2 receptor L) pyy.S (peptide YY S) pth2R.L (nociceptin receptor-like S) sstr5.S (somatostatin receptor 5 S) nmur1.L (neuromedin U receptor 1 L) grpr.L (gastrin releasing peptide receptor L) p2rx5.L (purinergic receptor P2X, ligand gated ion channel, 5 L) gpr83.2.L (G protein-coupled receptor 83 L) s1pr5. L (sphingosine-1-phosphate receptor 5 L) avpr1a.L (arginine vasopressin receptor 1A L) nts.L (neurotensin L) drd1.S (dopamine receptor D1 S) chrna7.S (cholinergic receptor, nicotinic alpha 7 S) galr3.L/S (galanin receptor 3 L/S) adm.S (adrenomedullin S) thrb.L (thyroid hormone receptor, beta L)	vip.S (vasoactive intestinal peptide S) htr5.L (5-hydroxytryptamine (serotonin) receptor 5A, G protein-coupled L) mc5r.L (melanocortin 5 receptor L) tacr3.L (tachykinin receptor 3 L) calcr.S (calcitonin receptor S) lpar1.L (lysophosphatidic acid receptor 1 L) drd2.L/S (dopamine receptor D2 L/S) grik2 (glutamate receptor, ionotropic, kainate 2) crhr2.S (corticotropin releasing hormone receptor 2 L/S) cga.L/S (glycoprotein hormones, alpha polypeptide S) ednrb2.S (endothelin receptor B subtype 2 S) lpar1.S (lysophosphatidic acid receptor 1 S) trh.L (thyrotropin-releasing hormone L) adcyap1.L (adenylate cyclase activating polypeptide 1 (pituitary) L) ghr.L (growth hormone receptor L) aplnr.L (apelin receptor L) pdyn.L/S (prodynorphin L) s1pr1.L (sphingosine-1-phosphate receptor 1 L) penk.L (proenkephalin L) prl.1.S (prolactin, gene 1) tac1.L/S (tachykinin precursor 1 L/S)	Neuroactive ligand-receptor interaction

## Data Availability

All fastq files are available on the GEO database GSE223829.

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
