# Peer review of "A Mixture of Chemicals Found in Human Amniotic Fluid Disrupts Brain Gene Expression and Behavior in *Xenopus laevis"

_ijms, 2023, doi:10.3390/ijms24032588_

Round 1
Reviewer 1 Report
ijms-2075129
The manuscript by Leemans et al. provides very interesting results about a very important research topic. The impact of EDCs on brain development is unfortunately not well understood yet. This work provides important data to understand the processes driven by disrupted TH signaling. The manuscript is very well written and organized. The results are well described and support the conclusions. I only have minor suggestions for improvement of this already excellent manuscript:
Lines 12-16: I think a part of the sentence missing?
Line 76: please provide a short statement regarding survival rates, growth effects and potential malformations of the animals.
Results paragraph: there is some inconsistency regarding the use of the abbreviation for “thyroid hormone”. Sometimes it is fully written, sometimes only “TH”.
Figure 2, 5 and 6: it is difficult to read the text in the figures. They should be be larger. I assume this will be changed in the editorial formatting process.
Table 2: These tables are very large and need to be read in detail to get the important information. It is a matter of taste, but there are more elegant ways to visualize pathways. E.g. http://cbl-gorilla.cs.technion.ac.il
Figure 5 has asterisks to display statistical differences, all other figures show the p-values.
Figure 6: the brain sections look different, for example, the area of the T4 sections is bigger. Also the structure differs. In M&M it was described that sections were taken at the same level for each animal. So is the difference we see here and effect of the exposure? It could be interesting to measure the area of the sections and then calculate the number of cells in relation to the area. Maybe the increase that is seen for the T4 animals will be less strong then. In any case, this morphological change is also a very relevant effect to report.
Line 154: a bit more details about the behavioral effects would be interesting. How strong was the decrease?
Line 311: maybe add a short sentence explaining on what the choice of concentrations was based?
Line 313: maybe not everyone knows what Evian is
Chemical exposure paragraph: please add more details regarding the experimental set up. How many animals per well? How many replicate runs for each experiment? How many concentrations, etc. Also in the following sections, it is not always clear how many animals were used for each measurement, e.g. the immunohistology
Why are the plates kept in the dark during the exposure? Is this standard? It can be assumed that this has an impact on the development of the animals.
Author Response
The manuscript by Leemans et al. provides very interesting results about a very important research topic. The impact of EDCs on brain development is unfortunately not well understood yet. This work provides important data to understand the processes driven by disrupted TH signaling. The manuscript is very well written and organized. The results are well described and support the conclusions. I only have minor suggestions for improvement of this already excellent manuscript.
We would like to thank the reviewer for taking the time and effort necessary to review the manuscript. We sincerely appreciate all valuable comments and suggestions, which helped us to improve majorly the quality of the manuscript. Our responses are given in a point-by-point manner below. Changes to the manuscripts are shown in underlined/red (tracked changes).
- Lines 12-16: I think a part of the sentence missing?
Line 17-19: The phrase has been corrected.
- Line 76: please provide a short statement regarding survival rates, growth effects and potential malformations of the animals.
Line 99 -103 The information on the survival rates, growth effects and potential malformations have been added to the manuscript.
- Line Results paragraph: there is some inconsistency regarding the use of the abbreviation for “thyroid hormone”. Sometimes it is fully written, sometimes only “TH”.
This inconsistency has been removed from the updated version of the manuscript.
- Figure 2, 5 and 6: it is difficult to read the text in the figures. They should be be larger. I assume this will be changed in the editorial formatting process.
Once we get to the editorial formatting process, high quality graphics and images will be provided. We thank the reviewer for indicating this.
- Table 2: These tables are very large and need to be read in detail to get the important information. It is a matter of taste, but there are more elegant ways to visualize pathways. E.g. http://cbl-gorilla.cs.technion.ac.il
Unfortunately, this gene ontology enrichment tool does not include the organism of interest, Xenopus laevis. Therefore, we opted to update the table to a more readable format.
- Figure 5 has asterisks to display statistical differences, all other figures show the p-values.
This inconsistency has been corrected in the updated version of the manuscript.
- Figure 6: the brain sections look different, for example, the area of the T4 sections is bigger. Also the structure differs. In M&M it was described that sections were taken at the same level for each animal. So is the difference we see here and effect of the exposure? It could be interesting to measure the area of the sections and then calculate the number of cells in relation to the area. Maybe the increase that is seen for the T4 animals will be less strong then. In any case, this morphological change is also a very relevant effect to report.
This a very valuable comment and we thank the reviewer for pointing this out. Within our set of represented pictures, we chose three images with each image corresponding to one exposure which were 1/ within a similar brain region and 2/representing a visible marking of both markers (Caspase3 and PH3). A more similar region of T4 did not contain Caspase3 marking and was therefore not selected for the representation within the paper (figure attached below). Measurement of both markers (PH3 and Casp3) was conducted over the whole brain. Furthermore, it should be noted that T4 was included as a positive control within this paper. It is well documented that T4 increases cell proliferation and differentiation (Cline et al. 2016 – DOI: 10.1523/JNEUROSCI.4147-15.2016). Results of immunohistochemistry on T4-exposed animals confirm these results. This information has now been added to the manuscript.
- Line 154: a bit more details about the behavioral effects would be interesting. How strong was the decrease?
Information about the behavioral effects has been added to the manuscript: The distance traveled decreased with the mixture- and TH-exposure in the light periods with 32% on average for the 1x mixture-treated animals and with 38% for the TH-treated animals (line 187-188).
- Line 311: maybe add a short sentence explaining on what the choice of concentrations was based?
Line 75-90 and line 23: The choice of the concentrations was based on real-life exposure measurements and has been thoroughly explained in the following article: Fini et al., 2017 ‘ Human amniotic fluid contaminants alter thyroid hormone signalling and early brain development in Xenopus embryos’. The text has been adapted to make the origin of the concentration clearer.
- Line 313: maybe not everyone knows what Evian is
We thank the reviewer for pointing this out and clear referencing has been employed.
- Chemical exposure paragraph: please add more details regarding the experimental set up. How many animals per well? How many replicate runs for each experiment? How many concentrations, etc. Also in the following sections, it is not always clear how many animals were used for each measurement, e.g. the immunohistology
Details about replicate runs and animals has been added after each endpoint described in the material and methods section (line 304-307, line 445-447, line 473-475, line 492-495, line 515-516)
- Why are the plates kept in the dark during the exposure? Is this standard? It can be assumed that this has an impact on the development of the animals.
Multi-well plates were kept in the dark incubator at 23°C for eight days to prevent chemical and hormones from degradation related to light exposure. This information has been added on lines 411 – 412.
Reviewer 2 Report
General comment
- The title "A mixture of chemicals found in human amniotic fluid disrupts 2 brain gene expression and behavior in Xenopus laevis" is very interesting that gives more information about the fields, as well as demonstrating the correlation between thyroid hormone and brain development, which is particularly fascinating in the field of brain research in relation to molecular toxicology.
- The abstract, introduction, result and discussion as well as methods are well described
Minor comments
- Line 88 Fig F should be Fig 2F
- The font type for Figure 1 should be changed
- Change Figure 4A font type
- All figures should present significant values in the same way. For instance, on some figures the precise p-value is mentioned (Fig.2, 6b), but on others Asterix is used (Fig. 5)
- The reference style should be based on the journal reference style format- must be in numbers
Author Response
General comment
- The title "A mixture of chemicals found in human amniotic fluid disrupts 2 brain gene expression and behavior in Xenopus laevis" is very interesting that gives more information about the fields, as well as demonstrating the correlation between thyroid hormone and brain development, which is particularly fascinating in the field of brain research in relation to molecular toxicology.
- The abstract, introduction, result and discussion as well as methods are well described
We would like to thank the reviewer for his positive/her positive response. We corrected the pointed-out mistakes within the updated version of the manuscript which can be identified in underline/red/bold.
Minor comments
- Line 88 Fig F should be Fig 2F
This mistake has been corrected in the updated version of the manuscript.
- The font type for Figure 1 should be changed
The font type has been altered to correspond to the rest of the text within the figure.
- Change Figure 4A font type
The font type has been modified to correspond with the rest of the text within the figure.
- All figures should present significant values in the same way. For instance, on some figures the precise p-value is mentioned (Fig.2, 6b), but on others Asterix is used (Fig. 5)
This inconsistency has been corrected in the updated version of the manuscript.
- The reference style should be based on the journal reference style format- must be in numbers
The reference style has been adapted to the journals’ reference style format.